# A Graph-Based Approach to Identify Factors Contributing to Postoperative Lung Cancer Recurrence among Patients with Non-Small-Cell Lung Cancer

**DOI:** 10.3390/cancers15133472

**Published:** 2023-07-03

**Authors:** Kartik Iyer, Shangsi Ren, Lucy Pu, Summer Mazur, Xiaoyan Zhao, Rajeev Dhupar, Jiantao Pu

**Affiliations:** 1Department of Radiology, University of Pittsburgh, Pittsburgh, PA 15213, USA; iyer.kartik@medstudent.pitt.edu (K.I.); renshangsi@gmail.com (S.R.); xiz318@pitt.edu (X.Z.); 2Department of Cardiothoracic Surgery, Division of Thoracic and Foregut Surgery, University of Pittsburgh, Pittsburgh, PA 15213, USA; lucyirann05@gmail.com (L.P.); mazursn2@upmc.edu (S.M.); dhuparr2@upmc.edu (R.D.); 3Surgical Services Division, Thoracic Surgery, VA Pittsburgh Healthcare System, Pittsburgh, PA 15213, USA; 4Department of Bioengineering, University of Pittsburgh, Pittsburgh, PA 15213, USA

**Keywords:** causal graph, non-small-cell lung cancer, recurrence, recurrence-free survival, body composition

## Abstract

**Simple Summary:**

This study aimed to model the causal relationship between radiographical features derived from CT scans and postoperative lung cancer recurrence and recurrence-free survival. A cohort of 363 lung cancer patients was retrospectively identified, and a novel causal graphical model was used to identify and visualize the causal relationship between these factors. Body composition, particularly with respect to adipose tissue distribution, was found to have a significant and causal impact on both recurrence and recurrence-free survival.

**Abstract:**

The accurate identification of the preoperative factors impacting postoperative cancer recurrence is crucial for optimizing neoadjuvant and adjuvant therapies and guiding follow-up treatment plans. We modeled the causal relationship between radiographical features derived from CT scans and the clinicopathologic factors associated with postoperative lung cancer recurrence and recurrence-free survival. A retrospective cohort of 363 non-small-cell lung cancer (NSCLC) patients who underwent lung resections with a minimum 5-year follow-up was analyzed. Body composition tissues and tumor features were quantified based on preoperative whole-body CT scans (acquired as a component of PET-CT scans) and chest CT scans, respectively. A novel causal graphical model was used to visualize the causal relationship between these factors. Variables were assessed using the intervention do-calculus adjustment (IDA) score. Direct predictors for recurrence-free survival included smoking history, T-stage, height, and intramuscular fat mass. Subcutaneous fat mass, visceral fat volume, and bone mass exerted the greatest influence on the model. For recurrence, the most significant variables were visceral fat volume, subcutaneous fat volume, and bone mass. Pathologic variables contributed to the recurrence model, with bone mass, TNM stage, and weight being the most important. Body composition, particularly adipose tissue distribution, significantly and causally impacted both recurrence and recurrence-free survival through interconnected relationships with other variables.

## 1. Introduction

Lung cancer is the leading cause of cancer-related death. With a low five-year mortality rate of 15.6%, it accounted for nearly 1.76 million deaths in 2020 [1]. Non-small-cell lung cancer (NSCLC) accounts for more than 80% of lung malignancies, and surgical resection is considered the primary curative option [2]. However, even after successful surgical resection, the cancer recurrence rate remains alarmingly high, ranging from 30% to 55% [3,4], leading to poor outcomes [5,6]. Although follow-up computed tomography (CT) imaging after resection is crucial in identifying most cases of recurrence, current postoperative surveillance strategies do not consider individual risk factors beyond the pathologic stage [7,8]. Identifying additional preoperative factors that can influence postoperative cancer recurrence would allow for the better assessment of the likelihood of recurrence, which would be valuable in facilitating neoadjuvant and adjuvant therapy [9,10] to minimize recurrence and guiding initial follow-up treatment plans following surgical resection.

The available studies have shown that several factors are linked to postoperative lung cancer recurrence, including patient-related factors such as smoking, age, or gender [11,12,13,14]; perioperative factors such as surgical trauma, transfusion, hypothermia, and anesthesia [15,16,17]; systemic inflammation [18]; molecular biomarkers [19,20,21]; and image features like tumor and body composition features [22,23,24]. While most of these studies only focused on a limited number of variables, the complex pathophysiology of NSCLC means that many variables may interact with each other with regard to contributing to recurrence. There is, therefore, a need to develop a causal model to understand not just the correlations but the potential causal relationships between the relevant variables that may be associated with lung cancer recurrence. This is especially important for body composition variables, which have been linked to a number of other diseases, including cardiovascular disease [25] and diabetes [26].

Causal artificial intelligence (AI) is a valuable method with which to understand the underlying causal relationships between variables, particularly in the field of biomedicine and clinical medicine [27,28,29]. This method involves utilizing the current values of various variables to perform a backward prediction to infer the relationship each variable has with all others and with the outcome of interest (e.g., recurrence, overall survival, etc.). While the traditional Greedy Equivalence Search (GES) [30] can generate a model with an optimized score, it treats all variables, including outcome variables, equally in terms of being possible predictors. As a result, a downside of GES is that it will present causations and relationships that defy common sense, such as overall survival predicting tumor stage instead of vice versa. To address this limitation, we recently developed a novel method, termed “*Grouped* Greedy Equivalence Search” (GGES), which allows for the classification of variables as predictors and outcome variables, thus generating a directed causal graph with more sensible relationships. 

The purpose of this study Is to explore and understand the causal relationship between various preoperative factors and two key outcomes, namely, (1) recurrence and (2) recurrence-free survival (RFS), among patients with NSCLS who have undergone surgical resection. This study utilized a cohort of NSCLS patients with a minimum follow-up of five years and incorporated preoperative chest CT and PET-CT scans to measure tumor characteristics and body composition. Causal models were constructed to predict either recurrence or RFS. The impact of each variable on the model was calculated to determine the most crucial predictive factors for each endpoint. To the best of our knowledge, this is the first use of the causal discovery model or causal AI, specifically the GGES method, to investigate postoperative recurrence among NSCLS patients. The results of this study may shed light on the complex interactions between predictive factors and highlight important variables for further investigation. 

## 2. Materials and Methods

### 2.1. Study Population

A cohort was identified from an active lung cancer patient database maintained at our medical center. The inclusion criteria were as follows: (1) NSCLC diagnosis, (2) availability of preoperative chest CT and PET-CT scans (from neck to thigh), (3) curative intent lung resection, (4) at least 5 years of follow-up history, and (5) no preoperative chemotherapy, immunotherapy, clinical trial, or radiation. Chest CT scans with a slice thickness greater than 2.5 mm were excluded from the study. The pre-treatment chest CT and PET-CT scans closest to the surgery date were used to quantify body composition and tumor characteristics. CT scans acquired as part of the PET-CT scans conducted on the neck to the mid-thigh were referred to as whole-body CT scans and were used to quantify body composition. A total of 363 patients that met the criteria were identified (Table 1). Subject information was de-identified and re-identified with a unique study ID number by an honest broker. This study was approved by the University of Pittsburgh Institutional Review Board (IRB) (IRB #: STUDY20100305).

### 2.2. Image Acquisition

The chest CT examinations in our cohort spanned more than ten years and were acquired using different protocols and scanners. The CT scanners included Optima-CT660, LightSpeed-VCT, LightSpeed-Ultra, Emotion, and Emotion-Duo. All chest CT scans were conducted (without using a radiopaque contrast agent) on the participants in a supine position while they held their breath at the end of inspiration. Images were reconstructed to encompass the entire lung field in a 512 × 512-pixel matrix using different reconstruction kernels. The in-plane pixel dimensions ranged from 0.55 to 0.82 mm, and the slice thickness ranged from 0.625 to 2.5 mm.

### 2.3. Study Variables 

(1)Clinicopathologic features: Clinicodemographic information included age, gender, race, weight, height, smoking status, and surgery details. Race was coded as follows: (1) white, (2) African American, or (3) other. Smoking status was coded as follows: (1) current and prior smokers or (2) no smoking history. The histopathological information included pathological TNM staging and histopathologic subtypes (HPS): (1) adenocarcinoma, (2) squamous cell carcinoma, or (3) other.(2)Body composition tissues depicted on whole-body CT scans: We developed a convolutional-neural-network (CNN)-based deep learning algorithm to automatically segment five different body tissues depicted on the CT images, including visceral adipose tissue (VAT), subcutaneous adipose tissue (SAT), intermuscular adipose tissue (IMAT), skeletal muscle (SM), and bones [31]. We used this algorithm to identify these body tissues on the whole-body CT scans obtained as part of PET-CT examinations. Compared to chest CT scans, whole-body CT scans enable a more accurate assessment of body composition [32]. Based on the segmentation, volume and mean density (i.e., average Hounsfield (HU) value) were computed for each body tissue.(3)Tumor features based on dedicated chest CT scans: Lung tumors in the cohort were automatically segmented using our available algorithm [33], and 10 CT image features were quantified: (1) volume, (2) mean density, (3) surface area, (4) maximum diameter, (5) mean diameter, (6) solidness, (7) mean diameter of the solid part, (8) cavity ratio, (9) calcification volume, and (10) irregularity. We used a threshold of −300 HU to determine the solid component of a nodule. A threshold of −910 HU was used to determine the cavitation within a nodule. The irregularity of a nodule was calculated as the ratio between its surface area and volume. The calcification volume was computed as the volume in the nodule with a density greater than 200 HU.

Before the causal analysis, all non-numeric variables were encoded into categorical variables, while numeric variables were passed through without encoding.

### 2.4. Causal Discovery Modeling Based on Grouped Greedy Equivalence Search (GGES)

The process of causal discovery aims to uncover the underlying causal relationships between variables in a dataset by using data as evidence. However, traditional methods for causal discovery have strict requirements for the dataset and the variables [34], including large sample sizes and complete information about the variables. These requirements limit the applicability of these methods to real-world datasets. This often results in inaccurate conclusions or illogical causal relationships. To address these limitations, we proposed a new method called the GGES. GGES leverages prior knowledge to facilitate the identification of the most likely relationships between variables in a dataset and produces a local optimal solution. An overview of the methodology is provided in Figure 1. 

The GGES method is a score-based method [35] derived from the GES Method [30]. The GGES method requires that the employed dataset satisfies the Markov condition [36] and the faithfulness hypothesis [37]. In the GGES method, the score of the graphical model can be calculated using the Bayesian Information Criterion (BIC) score [38] and a generalized score [39]. The GGES method has three steps:
(1)Group Process: All variables are divided into two groups: the causal variable group (Vc) and the predicted variable group (Vp). The groups have the following constraint condition (C) (Equation (2)): the predicted variables are assumed to have no influence on other variables within their group and are only influenced by other variables. On the other hand, causal variables can both cause and be influenced by other variables. This grouping process requires some prior knowledge about which variables represent the outcomes. (2)Greedy Forward Search (GFS) Process: The GGES method starts with an initial empty graph and sets the score of the whole graph to 0. Then, the method proceeds with adding edges between variables in a sequential manner, calculating the graph score after each addition and comparing the scores to select the model with the highest score. Nodes assigned to the outcome variable group are skipped according to the constraint placed at the beginning of the process (Equation (1)).(3)Greedy Backward Search (GBS) Process: The GGES then performs a backward search, where each edge is deleted sequentially from the selected graph model, and a new graph model is calculated. The graph model with the highest graph score is selected as the final causal graph result, which represents the inferred causal relationship between the variables in the dataset (Equation (1)):

(1)$$\max\;f\left(G,D\right)\;s.t.G\in \Omega ,G|=C$$(2)$$C\left\{\begin{array}{cc} v_{i}\to v_{j}, & v_{i}\epsilon Vc,v_{j}\epsilon Vp\\ v_{j}↛v_{i}, & v_{i}\epsilon Vc,v_{j}\epsilon Vp\\ v_{i}\to v_{j}, & {(v}_{i},v_{j})\epsilon Vc\\ v_{i}↛v_{j}, & {(v}_{i},v_{j})\epsilon Vp \end{array}\right.$$
where f is the structure scoring function, Ω is the structure space, and G|=C means that G satisfies the constraint condition C. In the process of searching and scoring, the constraint condition C is implemented to require the searched structure to satisfy the acyclic structure in the structure graph.

### 2.5. Training GGES Models 

The GGES model was applied to explore the causal relationship between two sets of variables: (1) all variables and *REC_FREE_SURVIVAL_(months)* and (2) all variables and recurrence (*Recur*). Then, variables that were directly related to the *REC_FREE_SURVIVAL_(months)* and *Recur* relationship were examined independently to determine the causality of the variables that contributed the most to each stage. The final model combines the independent analyses of each component with an overall analysis to identify the most appropriate causal model. The contribution of each variable was evaluated using the intervention do-calculus adjustment (IDA) score [40].

### 2.6. Variable Selection

The data cohort includes 119 variables extracted from 363 patients. When building causal graphs, all variables were initially included; any variables that had no connections to any other variables within a graph were considered non-significant and thus removed. After removing non-significant variables, 70 variables were identified from our dataset, namely, 15 body composition variables and 5 body composition ratio variables, 7 demographic variables, 12 surgery-related variables, 15 vessel-related variables, 5 death-timing-related variables, 2 pathologic stage variables, and 5 other variables. 

### 2.7. Performance Validation

The 10-fold cross-validation method was used to train and validate the models with respect to the study cohort to predict (1) *REC_FREE_SURVIVAL_(months)* and (2) *Recur*. To determine the causal relationships between variables, the training dataset was fed into the causal discovery model to generate a causal graph. The test dataset was used to evaluate the causal results based on AP, AR, AHP, and AHR [41]. The Total Causal Effects (TCE) value [42] was calculated to determine the strength of the relationship between variables, where a higher absolute TCE value indicates a stronger relationship. All statistics were performed using R 3.4.1 and Python. A *p*-value less than 0.05 was considered statistically significant.

## 3. Results

### 3.1. Causal Analysis of Recurrence-Free Survival

Based on the generated causal graph, 22 variables were identified as having a relationship with recurrence-free survival (Figure 2). The variables *Smoke_Hx_Coded, T_size, height*, and *mass_intermuscular_fat* were directly linked to *Rec_Free_Survival_(months)*. Three variables demonstrated significantly higher IDA scores compared to the others, namely, *mass_subcutaneous_fat, volume_visceral_fat*, and *mass_bone*, all displaying |IDA| values exceeding 40. Table 2 provides a comprehensive overview of the IDA scores between the variables and *Rec_Free_Survival_(months)*.

### 3.2. Causal Analysis of Recurrence

First, a causal graphical model was constructed using solely demographic and body composition variables. In total, sixteen variables were found to have causal associations with recurrence (Figure 3). The node directly linked to *Recur* was *density_muscle*, while the only two nodes directly connected to *TNM_stage* were *n_stage* and *t_stage*. Among the variables, *volume_visceral_fat* displayed the highest IDA score (6.3379), followed by *volume_subcutaneous_fat* and *mass_bone* as the next two variables with the highest IDA scores. Table 3 showed a comprehensive list of the identified variables and their IDA scores. 

Next, a comprehensive causal model was generated by combining three categories of variables: demographics, body composition tissues, and pathologic TNM stage. Among these variables, sixteen variables were found to have causal associations with recurrence (Figure 4). The node directly linked to *Recur* was *TNM_stage*, while the only two nodes directly connected to *TNM_stage* were *n_stage* and *t_stage*. In the model, the variable with the highest IDA score was *mass_bone*, scoring 0.29. Following *mass_bone*, the next two variables with the highest scores were *TNM_stage* and *weight*. Table 4 showed a comprehensive list of the identified variables and their IDA scores.

## 4. Discussion

This study highlights the usefulness of the GGES method for discovering the key factors affecting recurrence and recurrence-free survival (RFS) among NSCLC patients after surgical resection. Radiomic and tumor features were automatically computed from preoperative CT images, while whole-body CT scans (from neck to thigh) were used to segment and quantify five different tissues related to body composition. The key predictors of RFS were found to be weight, T-stage, smoking history, and intramuscular fat mass, with subcutaneous fat mass, visceral fat volume, and bone mass bone having the most significant impact on the model. Regarding recurrence, the variables with the greatest influence were found to be visceral fat volume, subcutaneous fat volume, and bone mass.

Among all the variable categories considered, body composition variables were the most significant predictors of outcomes for both RFS and recurrence. The highest IDA scores were assigned to the variables of visceral and subcutaneous adipose tissues and bone mass. This suggests that visceral and subcutaneous adipose tissues depicted on CT images play a crucial role in predicting recurrence and RFS but also exert their effects through connections with other, less-visible features. These findings align with those of previous studies that reported on the negative impact of visceral fat on survival among cancer patients [43,44].

The role of adipose tissue in cancer prognosis remains a topic of discussion. On the one hand, there is the well-known obesity paradox, wherein increased levels of adipose tissue are associated with the increased survival of patients with NSCLC [45]. On the other hand, an increased BMI is known to increase mortality for many types of malignancies, which is possibly due to the upregulation of growth hormones like IGF-1 caused by the increased amount of adipose tissue [46]. The results of this study showed that increased weight caused decreased recurrence, but increased components of subcutaneous and visceral fat caused increased recurrence and decreased RFS. This supports the hypothesis that the distribution of adipose tissue is more crucial in this respect than overall BMI [47]. 

It is also noteworthy that bone mass ranked among the top three variables contributing to all three models. The significance of bone mass and surrogates, particularly bone mineral density (BMD), has been demonstrated to have prognostic value for diseases like cardiovascular disease and chronic lung disease [48] as well as many cancers like breast cancer [49], colorectal cancer [50], and, importantly, NSCLC [51]. While the exact mechanism of BMD’s influence is unclear, it has been suggested that BMD may reflect tumor-induced metabolic and hormonal changes. The causal models generated in this study suggest that BMD’s effect is manifested through its influence on other variables, notably adipose tissue distribution, before finally affecting recurrence and survival. Moreover, as gender is recognized as a factor influencing BMD, it is conceivable that gender disparities could also contribute to hormone-related recurrence outcomes. Further comprehensive investigations are warranted in order to delve deeper into these causal relationships. 

When considering the variables directly linked to lung cancer recurrence, a notable difference was observed in the graphs when pathological stage variables were included. Previously, the direct predictor was *density_muscle*, but upon inclusion of pathological stage variables, TNM-stage emerged as the direct predictor, followed by T-stage and N-stage. This finding aligns with the well-established understanding that the overall stage, particularly the N-stage, significantly impacts recurrence. Furthermore, the introduction of stage variables resulted in a reduction in the number of significant variables contributing to predicting recurrence. This phenomenon may be attributed to the stronger correlation between body composition variables and stage variables. Consequently, variables such as T-stage alone may not be sufficient to explain the recurrence phenomenon. These findings provide additional evidence supporting the interplay between body composition variables and other extensively studied predictors of recurrence and RFS.

This study has several limitations that should be considered. One is the heterogeneity of the CT-scanning protocols, which might have introduced variations in the values of the radiomics features due to the differences in CT voxels. Despite this challenge, this study showed the potential of body composition variables as predictors of recurrence and RFS. Another limitation is that the GGES method requires some basic knowledge of the relationship between variables, although this method reduces the number of unreasonable causal relationships drawn. Thus, with further research, this information can be refined to produce improved predictive graphs. In addition, this study did not incorporate postoperative complications into its models. While we did not have access to these data, we believe our models could be improved with the addition of such variables. Finally, to minimize the influence of potential confounding factors on our analysis, we specifically collected a cohort of patients who did not receive neoadjuvant therapy. However, this approach resulted in a failure to account for adjuvant therapies, which evidently had an impact on the outcomes. Two recent phase 3 trials [9,10] showed that the addition of pembrolizumab (Keytruda) to neoadjuvant platinum-based chemotherapy followed by resection and adjuvant pembrolizumab alone and neoadjuvant nivolumab plus chemotherapy resulted in a significant improvement in event-free survival (EFS) and pathological response for patients with early-stage NSCLC. Hence, we will incorporate this information in future studies. 

## 5. Conclusions

Our graph-based causal analyses showed that body composition features, especially those related to visceral fat, subcutaneous fat, and bone, presented the highest degree of contribution to the models, even when other factors like T-stage were direct predictors. These models highlight the importance of body composition as an underlying mechanism behind recurrence and RFS, while the causal relationships contained within suggest new avenues for exploring relationships between variables not previously understood. 

## Figures and Tables

**Figure 1 cancers-15-03472-f001:**
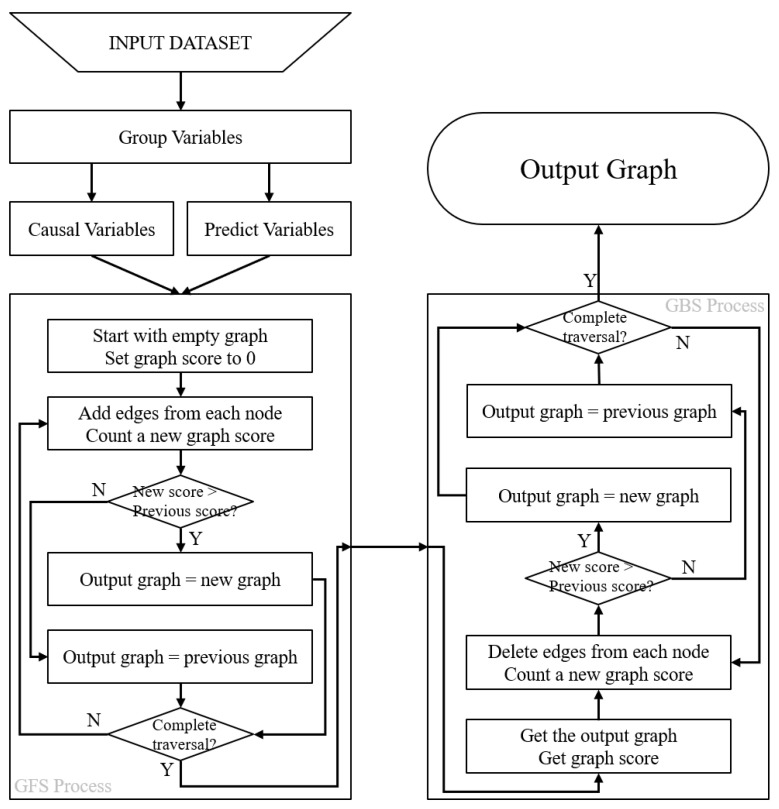
Illustration of the Grouped Greedy Equivalence Search (GGES) method. The process begins with the division of the variables into two groups, namely, the causal variables group and the outcome variables group, with an initial graph score of 0. The left panel depicts the Greedy Forward Search (GFS), where edges are incrementally added, and the new graph score is updated. The graph with the highest score is retained at each step. The right panel shows the Greedy Backward Search (GBS), where edges are removed and the new graph score is recalculated, for which the graph with the higher score is kept once again. The final causal graph is obtained after the GFS and GBS steps are completed.

**Figure 2 cancers-15-03472-f002:**
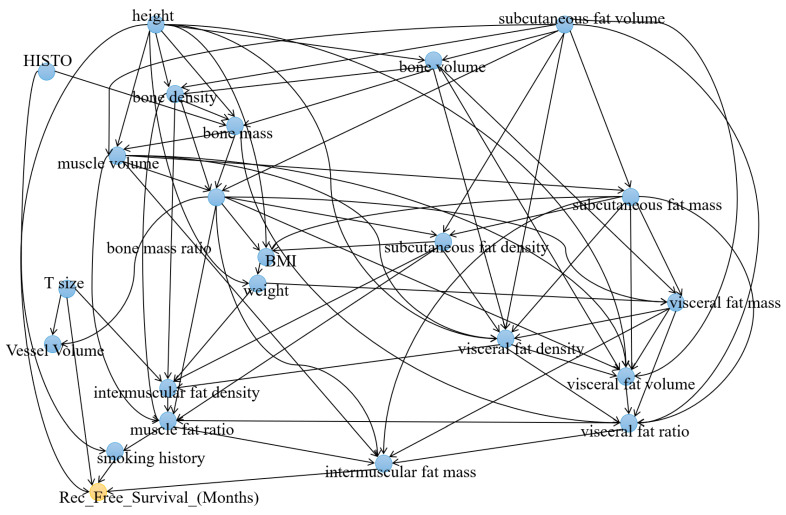
Causal graph between all groups of variables and recurrence-free survival (*REC_FREE_SURVIVAL_(months)*).

**Figure 3 cancers-15-03472-f003:**
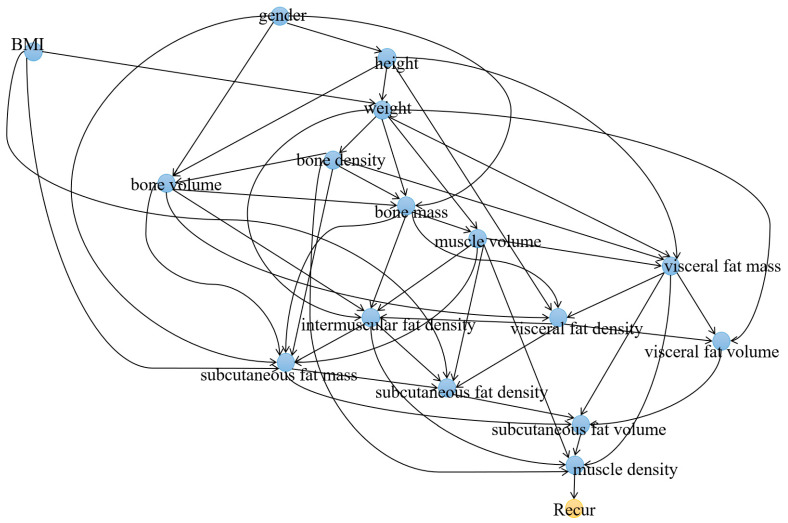
Causal graph depicting the relationship between demographic and body composition variables and lung cancer recurrence (*Recur*).

**Figure 4 cancers-15-03472-f004:**
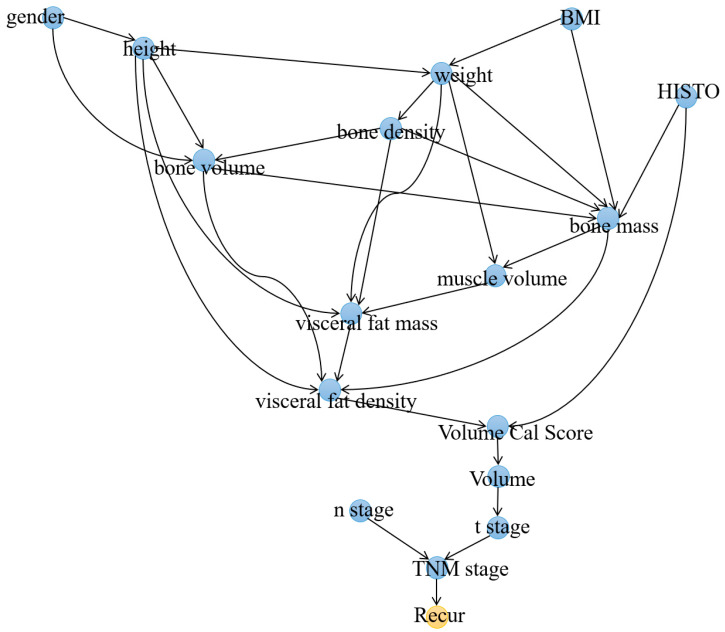
Causal graph depicting the relationship between demographic, body composition, and pathologic stage variables and lung cancer recurrence (*Recur*).

**Table 1 cancers-15-03472-t001:** Subject demographics in our cohort (*n* = 363).

Characteristics	Value ^1^
Age	68.3 ± 9.45
Height (cm)	174.4 ± 43.0
Weight (kg)	66.3 ± 4.02
BMI	27.8 ± 5.98
Sex	
Female	168 (46.28%)
Male	195 (53.72%)
Race	
White	326 (89.81%)
Black	34 (9.37%)
Asian	3 (0.82%)
Surgical method	
lobectomy	292 (80.44%)
Segmentectomy and wedge resection	62 (17.08%)
Pneumonectomy	9 (2.48%)
Tumor site	
RUL	147 (40.50%)
RML	26 (7.16%)
RLL	50 (13.77%)
LUL	91 (25.07%)
LLL	49 (13.50%)
Overall pathological stage	
0 (NED)	6 (1.65%)
1A1, 1A2, 1A3, 1B	188 (51.79%)
2A, B	103 (28.37%)
3	66 (18.18%)
T stage	
0	6 (1.65%)
1A, B, C	138 (38.02%)
2A, B	157 (43.25%)
3	47 (12.95%)
4	15 (4.13%)
N stage	
0	252 (69.42%)
1	68 (18.73%)
2	43 (11.85%)
Recurrence	0.67 ± 0.47
Rec_free_survival_(months)	12.53 ± 16.29

^1^ Mean ± (SD); *n* (%).

**Table 2 cancers-15-03472-t002:** Variables and their IDA scores included in the recurrence-free survival model. (IDA: intervention do-calculus adjustment. BMI: body mass index).

Variables	IDA Score	Variables	IDA Score
density_intermuscular_fat	−0.008	vis_fat_ratio	1.569
Weight	−0.033	mass_intermuscular_fat	−1.600
volume_muscle	0.231	height	1.696
Vessel_Volume.ml.	−0.299	HISTO_CODED	−1.822
density_subcutaneous_fat	−0.314	Tsize	−2.371
BMI	−0.384	volume_bone	−2.421
volume_subcutaneous_fat	−0.740	SMOKE_HX_CODED	−3.106
density_bone	0.798	muscle_fat_ratio	−3.776
bone_mass_ratio	0.964	mass_bone	41.781
density_visceral_fat	1.121	volume_visceral_fat	−43.176
mass_visceral_fat	−1.224	mass_subcutaneous_fat	−76.002

**Table 3 cancers-15-03472-t003:** Variables and their IDA scores included in the recurrence causal graphical model based on subject demographics and body composition features. (IDA: intervention do-calculus adjustment. BMI: body mass index).

Variables	IDA Score
*density_visceral_fat*	0.002
*density_bone*	0.004
*mass_subcutaneous_fat*	−0.010
*density_subcutaneous_fat*	0.011
*density_intermuscular_fat*	0.017
*volume_bone*	−0.030
*BMI*	−0.032
*SEX*	0.033
*volume_muscle*	0.041
*height*	0.054
*mass_visceral_fat*	−0.062
*density_muscle*	0.127
*weight*	−0.237
*mass_bone*	0.665
*volume_subcutaneous_fat*	1.230
*volume_visceral_fat*	6.338

**Table 4 cancers-15-03472-t004:** Variables and their IDA scores in the recurrence causal graphical model based on demographics, body composition tissues, and pathologic TNM stage. (IDA: intervention do-calculus adjustment. BMI: body mass index).

Variables	IDA Score
*density_visceral_fat*	0.002
*density_bone*	0.004
*volume_bone*	−0.030
*BMI*	−0.032
*Gender*	0.033
*volume_muscle*	0.041
*Volume_Cal._Score.mm3.*	0.047
*HISTO_CODED*	0.048
*height*	0.054
*t_stage*	0.057
*mass_visceral_fat*	−0.062
*Volume.ml.*	0.065
*n_stage*	0.109
*weight*	−0.237
*TNM_stage*	0.281
*mass_bone*	0.291

## Data Availability

Data available upon request.

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
