# Peer review of "A Graph-Based Approach to Identify Factors Contributing to Postoperative Lung Cancer Recurrence among Patients with Non-Small-Cell Lung Cancer"

_cancers, 2023, doi:10.3390/cancers15133472_

Round 1

Reviewer 1 Report

Please add a STROBE checklist

Please improve the discussion.

Minor spelling errors

Author Response

Response to Referee #1’s Comments

  1. Please add a STROBE checklist

Answer: Per recommendation, we added a STROBE checklist.

  1. Please improve the discussion.

Answer: We improved the discussion as suggested.

  1. Minor spelling errors

Answer: We have proofread the manuscript to improve the writing.

Reviewer 2 Report

The article presented by the authors entitled "Causal graph modeling of factors associated with postoperative lung cancer recurrence in patients with non-small cell lung cancer" shows a study on the possible risk factors for recurrence of localised NSCLC based on a causal graph model. I believe that the research carried out has a novel and accurate approach to a very fashionable and current topic. Undoubtedly, it provides a series of new data on NSCLC that could have a great future impact on the follow-up of patients operated on for NSCLC, adapting this follow-up to their risk of recurrence. It could also have implications for future neoadjuvant treatments, optimising treatment regimens in cases with a higher risk of recurrence. Therefore, I believe the article could be of great interest for publication in the journal.

The manuscript needs a number of modifications and changes, which if done correctly would make the article publishable and of great interest to readers. The main changes that in my opinion the article needs are clarifications to make it easier to read and understand the results obtained (see below). On a general level, the article is written correctly, with the expressions and language being acceptable and not in need of major modifications. The structure is also correct, and the manuscript is easy to read, being possible to read it quickly if necessary. The figures and tables need some minor modification but are broadly appropriate. The references given are in line with the subject matter, however, in many cases they are old and need to be updated as the subject matter has a large amount of current literature. 

The following changes are proposed to the article:

Major changes

-     Material and methods: The follow-up period used was 6 years. It is important to indicate the reason for this figure, to explain it and to indicate why not 5 years like most studies in this regard.

-     Results: the main point that has caught my attention in the article is to know if data have been collected on adjuvant treatment after surgery. Knowing which patients were given post-surgery chemotherapy, radiotherapy, immunotherapy or EGFR-targeted treatments is essential to assess the results correctly. These data may have had a fundamental influence on the data obtained, and therefore I believe it is important to indicate and clarify this in both the results and the discussion.

-     Results: A point explaining the general characteristics of the sample is essential. It is not possible to write an article on a patient cohort of more than 600 patients without a point that correctly explains the characteristics of the sample.

-     Discussion: The impact of BMD has been analysed separately in women and men. It is likely that at the hormonal level there is a correlation with the likelihood of recurrence, so it would be interesting to comment on this.

Minor changes

-     Title: change "non-small cell lung cancer" to "non-small-cell lung cancer".

-     Keywords: add "body composition" and change "lung cancer" to "non-small-cell lung cancer".

-   Abstract: Put the data indicated in "Summary background data" before the objectives.

-     Introduction: as in the title, change "non-small cell lung cancer" to "non-small-cell lung cancer".

-   Introduction: given that the greatest applicability I see for the results of the article is the optimisation of preoperative treatment with the new neoadjuvant therapies, it would be appropriate to include a paragraph on this in the article. Studies such as CheckMate-816 with nivolumab or KEYNOTE-671 with pembrolizumab could be even more important in patients with a high risk of recurrence.

-     References: as above, please update the citations used.

-   Table 3: if this table could be changed and made more graphical it would be much more understandable and would help the reading of the article.

-     Conclusion: I think the first sentence is not necessary as it is not a conclusion as such.

Undoubtedly, the article by the authors is of great interest to the scientific community and provides very novel results. In my view, a number of major changes are necessary, but if the authors make these changes correctly, I believe the article is worthy of publication in the journal.

Author Response

Response to Referee #2’s Comments

The article presented by the authors entitled "Causal graph modeling of factors associated with postoperative lung cancer recurrence in patients with non-small cell lung cancer" shows a study on the possible risk factors for recurrence of localised NSCLC based on a causal graph model. I believe that the research carried out has a novel and accurate approach to a very fashionable and current topic. Undoubtedly, it provides a series of new data on NSCLC that could have a great future impact on the follow-up of patients operated on for NSCLC, adapting this follow-up to their risk of recurrence. It could also have implications for future neoadjuvant treatments, optimising treatment regimens in cases with a higher risk of recurrence. Therefore, I believe the article could be of great interest for publication in the journal.

The manuscript needs a number of modifications and changes, which if done correctly would make the article publishable and of great interest to readers. The main changes that in my opinion the article needs are clarifications to make it easier to read and understand the results obtained (see below). On a general level, the article is written correctly, with the expressions and language being acceptable and not in need of major modifications. The structure is also correct, and the manuscript is easy to read, being possible to read it quickly if necessary. The figures and tables need some minor modification but are broadly appropriate. The references given are in line with the subject matter, however, in many cases they are old and need to be updated as the subject matter has a large amount of current literature. 

Undoubtedly, the article by the authors is of great interest to the scientific community and provides very novel results. In my view, a number of major changes are necessary, but if the authors make these changes correctly, I believe the article is worthy of publication in the journal.

Major changes

  1. Material and methods: The follow-up period used was 6 years. It is important to indicate the reason for this figure, to explain it and to indicate why not 5 years like most studies in this regard.

Answer: We sincerely apologize for the imprecise statement. Initially, our inclusion criterion was set for a 5-year follow-up period. However, upon finalizing the data, we discovered that the minimum follow-up duration for recurrence-free survivors was actually six years. We corrected the statement in the revision.

  1. Results: the main point that has caught my attention in the article is to know if data have been collected on adjuvant treatment after surgery. Knowing which patients were given post-surgery chemotherapy, radiotherapy, immunotherapy or EGFR-targeted treatments is essential to assess the results correctly. These data may have had a fundamental influence on the data obtained, and therefore I believe it is important to indicate and clarify this in both the results and the discussion.

Answer: We appreciate the reviewer’s insight into this. It is true that neoadjuvant and adjuvant therapy will affect postoperative survival and recurrence. When we created the cohort, we excluded the cases that received preoperative chemotherapy, immunotherapy, clinical trial, or radiation. However, regrettably, we were unable to incorporate information regarding adjuvant therapies due to the difficulty of obtaining the necessary data pertaining to adjuvant therapies for all cases. As a result, we acknowledge this limitation in our study and hope to incorporate such information in future studies.

  1. Results: A point explaining the general characteristics of the sample is essential. It is not possible to write an article on a patient cohort of more than 600 patients without a point that correctly explains the characteristics of the sample.

Answer: We summarized the general characteristics of the lung patients in our cohort in Table 1. Thank you!

  1. Discussion: The impact of BMD has been analysed separately in women and men. It is likely that at the hormonal level there is a correlation with the likelihood of recurrence, so it would be interesting to comment on this.

Answer: We appreciate this insight and provided some discussion about this.

Minor changes

  1. Title: change "non-small cell lung cancer" to "non-small-cell lung cancer".

Answer: In most available literature and studies, “non-small cell lung cancer” is used. Examples can be found at the following links from NCI official website and PubMed publications:

  • https://www.cancer.gov/types/lung/patient/non-small-cell-lung-treatment-pdq
  • https://pubmed.ncbi.nlm.nih.gov/?term=non-small-cell+lung+cancer&sort=pubdate

Hence, we prefer to keep our original title.

  1. Keywords: add "body composition" and change "lung cancer" to "non-small-cell lung cancer".

Answer: We updated the keywords as suggested.

  1. Abstract: Put the data indicated in "Summary background data" before the objectives.

Answer: We improve the abstract with the incorporation of the reviewer’s comments.  

  1. Introduction: as in the title, change "non-small cell lung cancer" to "non-small-cell lung cancer".

Answer: Please see our above response.

  1. Introduction: given that the greatest applicability I see for the results of the article is the optimisation of preoperative treatment with the new neoadjuvant therapies, it would be appropriate to include a paragraph on this in the article. Studies such as CheckMate-816 with nivolumab or KEYNOTE-671 with pembrolizumab could be even more important in patients with a high risk of recurrence.

Answer: Per recommendation, We cited and discussed the two mentioned phase 3 clinical trials. 

  1. References: as above, please update the citations used.

Answer: The two papers published in New England Journal of Medicine were cited as suggested.

  1. Table 3: if this table could be changed and made more graphical it would be much more understandable and would help the reading of the article.

Answer: We improved the tables as suggested.

  1. Conclusion: I think the first sentence is not necessary as it is not a conclusion as such.

Answer: We improved the conclusion to make it concise and clear. 

Round 2

Reviewer 2 Report

ARTICLE REVIEW (2)

After performing the second review of the article, I believe that the authors have correctly made most of the requested changes. The authors are to be congratulated for the effort they have put into making all the changes correctly. The main flaw in the article is the lack of knowledge of adjuvant treatment, however, I believe the authors have solved it correctly by including this information in the discussion. In addition, the rest of the changes have been made with the exception of two that are still pending.

Table 1 needs to be modified as indicated in the previous review and more data needs to be added. In a study with 600 patients, the characteristics of the sample should be better described than the authors do. For example, data such as tumour histology or stage should be indicated here. This point should be expanded further. In addition, I think, as previously stated, it would be appropriate to change "non-small cell" to "non-small-cell".

I do not think that any changes beyond these two are necessary. Therefore, if the authors definitely make them, I think the article can be published in the journal.

Author Response

After performing the second review of the article, I believe that the authors have correctly made most of the requested changes. The authors are to be congratulated for the effort they have put into making all the changes correctly. The main flaw in the article is the lack of knowledge of adjuvant treatment, however, I believe the authors have solved it correctly by including this information in the discussion. In addition, the rest of the changes have been made with the exception of two that are still pending.

Table 1 needs to be modified as indicated in the previous review and more data needs to be added. In a study with 600 patients, the characteristics of the sample should be better described than the authors do. For example, data such as tumour histology or stage should be indicated here. This point should be expanded further. In addition, I think, as previously stated, it would be appropriate to change "non-small cell" to "non-small-cell".

I do not think that any changes beyond these two are necessary. Therefore, if the authors definitely make them, I think the article can be published in the journal.

Answer: We updated Table 1 with more information as suggested by the reviewer, such as pathological stage and surgery method. We also updated “non-small cell” to “non-small-cell.” Thank you!